# High Diagnostic Yield and Clinical Utility of Next-Generation Sequencing in Children with Epilepsy and Neurodevelopmental Delays: A Retrospective Study

**DOI:** 10.3390/ijms25179645

**Published:** 2024-09-06

**Authors:** Daniel Charouf, Derryl Miller, Laith Haddad, Fletcher A. White, Rose-Mary Boustany, Makram Obeid

**Affiliations:** 1Division of Child Neurology, Department of Pediatric and Adolescent Medicine, American University of Beirut Medical Center, Riad El Solh, Beirut P.O. Box 11-0236, Lebanon; daniel.charouf@outlook.com (D.C.); lh68@aub.edu.lb (L.H.); rb50@aub.edu.lb (R.-M.B.); 2Division of Child Neurology, Department of Neurology, Indiana University School of Medicine, Riley Hospital for Children, Indianapolis, IN 46202, USA; miller89@iu.edu; 3Stark Neurosciences Research Institute, Indiana University School of Medicine, Indianapolis, IN 46202, USA; fawhite@iu.edu; 4Department of Anesthesia, Indiana University School of Medicine, Indianapolis, IN 46202, USA; 5Department of Biochemistry, Faculty of Medicine, American University of Beirut, Beirut P.O. Box 11-0236, Lebanon

**Keywords:** epilepsy, next-generation sequencing, genetic yield, refractory seizures, neurodevelopmental delays

## Abstract

Advances in genetics led to the identification of hundreds of epilepsy-related genes, some of which are treatable with etiology-specific interventions. However, the diagnostic yield of next-generation sequencing (NGS) in unexplained epilepsy is highly variable (10–50%). We sought to determine the diagnostic yield and clinical utility of NGS in children with unexplained epilepsy that is accompanied by neurodevelopmental delays and/or is medically intractable. A 5-year retrospective review was conducted at the American University of Beirut Medical Center to identify children who underwent whole exome sequencing (WES) or whole genome sequencing (WGS). Data on patient demographics, neurodevelopment, seizures, and treatments were collected. Forty-nine children underwent NGS with an overall diagnostic rate of 68.9% (27/38 for WES, and 4/7 for WGS). Most children (42) had neurodevelopmental delays with (18) or without (24) refractory epilepsy, and only three had refractory epilepsy without delays. The diagnostic yield was 77.8% in consanguineous families (18), and 61.5% in non-consanguineous families (26); consanguinity information was not available for one family. Genetic test results led to anti-seizure medication optimization or dietary therapies in six children, with subsequent improvements in seizure control and neurodevelopmental trajectories. Not only is the diagnostic rate of NGS high in children with unexplained epilepsy and neurodevelopmental delays, but also genetic testing in this population may often lead to potentially life-altering interventions.

## 1. Introduction

Advances in genetic testing modalities in the past decade have remarkably expanded our understanding of genetic epilepsies with the identification of hundreds of epilepsy-associated genes [1,2,3,4]. While it is generally well accepted that unexplained epilepsy is likely, at least partly, genetic in nature, the yield of genetic testing remains highly variable [5] without a consensus on the type of genetic testing and the patient population that should be tested. Identifying a genetic etiology can have a major impact on the choice of anti-seizure medications (ASMs), particularly in epilepsies related to mutations in the *SCN2A*, *SCN8A*, *CDKL5*, and *POLG* genes [6,7,8,9]. Genetic testing may also reveal inherited metabolic epilepsies that are eminently treatable with specific supplements, vitamins, dietary restrictions, or the ketogenic diet. Such interventions can be highly effective in controlling seizures and improving the neurodevelopmental trajectory in conditions including but not limited to pyridoxine- and folate-dependent seizures, glucose transporter 1 deficiency syndrome, and biopterin, cerebral folate, serine, and biotinidase deficiencies [3,10,11]. Accurately diagnosing genetic conditions improves health surveillance in diseases with well-characterized comorbidities and can help in preventing disease reoccurrence in at-risk families and future generations via informed reproductive counseling.

Next-generation sequencing (NGS) may improve the diagnosis of genetic epilepsies, but its reported diagnostic yield varies widely from 10% to 50% for both whole exome sequencing (WES) [12,13] and whole genome sequencing (WGS) [14,15]. This variability is likely due to the heterogeneity of the electroclinical phenotype within a common phenotypic patient population [12,16,17], differences in testing methodologies such as using singleton WES instead of including both parents (trio testing) [18], and patient selection. Indeed, many studies report a diagnostic yield of NGS in patients who already had prior nonrevealing genetic testing including the use of gene panels [19,20]. Determining diagnostic yield and clinical utility of genetic testing is important to gauge clinical decision making and resource allocation. This is particularly true in developing countries such as Lebanon where such data are completely lacking. Because patients with an unfavorable clinical course are more likely to benefit from additional workup that might help in refining their clinical management, genetic testing is commonly offered to patients with unexplained epilepsy that is drug-resistant and/or accompanied by neurodevelopmental delays at the American University of Beirut Medical Center (AUBMC). In this 5-year retrospective chart review at AUBMC, we sought to determine whether first-line testing with NGS in this patient population resulted in a high diagnostic yield and clinical utility.

## 2. Results

Electronic medical chart review uncovered 45 children who underwent NGS for seizures with neurodevelopmental delays or for medically intractable epilepsy. The overall diagnostic rate was 68.9% (27/38 for WES, and 4/7 for WGS). Diagnoses consisted of developmental and epileptic encephalopathies (DEEs), metabolic conditions, and neurodegenerative diseases (Table 1). The mutations’ coordinates are listed in Table 2. The 31 diagnosed cases demonstrated pathogenic mutations in 28 genes or likely pathogenic variants in another 3 genes (*TUBA1A*, *AP4M1*, *ATP1A2*), which were deemed diagnostic by the ordering epileptologist/neurogeneticist. Seventeen patients carried heterozygous autosomal dominant defects, and 14 others carried homozygous autosomal recessive mutations. A single patient in this cohort was defined as having a variant of uncertain significance (VUS), which was not deemed as clinically relevant. This cohort consisted of 23 boys and 22 girls with a median age of 1 year and 4 months at the time of testing (range: 1 month to 12.5 years). There were five patients with neonatal-onset seizures, and all five had diagnostic NGS. Most patients (42 out of 45) had substantial neurodevelopmental delays. Global delays were present in 32 patients, and another 10 had significant motor and/or language delays. The diagnostic yield of NGS was 70.8% for patients with non-refractory epilepsy and delays (17/24) and 72.2% for those with medically intractable epilepsy and delays (13/18). Only three had refractory epilepsy without delays, of whom one had a genetic defect. Eighteen children were the product of consanguineous unions, 14 of whom had identifiable (77.8%) mutations including a single de novo defect. Genetic abnormalities were uncovered in 16 of 26 non-consanguineous families (61.5%) made up of 10 inherited and 6 de novo mutations. The presence or absence of consanguinity could not be confirmed for one patient with a de novo *TUBA1A* mutation.

A positive genetic test resulted in major potentially life-saving changes in the medical management of six children (Table 3). Etiology-specific treatments were initiated in two children, one with a *FOLR1* and another with a *QDPR* gene mutation that cause folate transporter and dihydropteridine reductase deficiencies, respectively. Both patients showed remarkable improvements in neurodevelopmental trajectories and seizure control after the initiation of targeted treatments. The child with *FOLR1* was started on oral folinic acid supplementation and then switched to the intravenous form, leading to the complete cessation of seizures. This child’s sibling was screened and had a similar mutation prompting the initiation of folinic acid supplementation. She remained symptom-free with an almost normal developmental trajectory except for mild autistic features as previously reported [10]. The child with a *QDPR* defect was started on oral tetrahydrobiopterin supplements (10 mg/kg/day), in addition to L-dopa (7 mg/kg/day), 5-hydroxytryptophan (5 mg/kg/day), and folinic acid (20 mg/day), given the known associated folinic acid and neurotransmitter deficiencies in this condition [3]. Treatment resulted in a sustained more than 50% reduction in seizure frequency at the last follow-up one-year post-therapy. Because GRIN2B encodes subunit 2 of the N-methyl-D-aspartate receptor (NMDAR), and the detected defect results in a gain of function [21], the child with this gene mutation was started on memantine, an NMDAR antagonist, given the emerging literature on its potential benefits [21,22]. However, the family was noncompliant with therapy, and the patient’s clinical status remained unchanged with frequent seizures. Three patients had sodium channelopathies, resulting in the optimization of ASM regimens based on genetic test results with a substantial improvement in seizure control. In one patient, an *SCN1A* mutation prompted switching from lamotrigine to levetiracetam (60 mg/kg/day) and the later addition of ethosuximide (20 mg/kg/day). The use of sodium channel blockers like lamotrigine can worsen seizures in patients with *SCN1A* mutations that lead to a loss of function in voltage-gated sodium channels [9]. Following the switch in ASMs, seizures remitted, and the child was seizure-free for 18 months at the last follow-up. Two other patients with *SCN8A* defects were started on carbamazepine (30 mg/kg/day). Patients with an *SCN8A* mutation-related gain of function in voltage-gated sodium channels benefit from sodium channel blockers like carbamazepine [9]. Seizure frequency decreased from multiple daily episodes to weekly seizures in one child, and from monthly seizures to seizure-freedom for 6 months prior to the last follow-up in the other patient.

Identified genetic defects prompted genetic counseling in a dedicated clinical visit to discuss genetic results. During this visit, prognosis of the disease and required specialist referrals were broached. In 23 families with inherited diseases, pre-pregnancy counseling regarding potential future siblings was performed. Fifteen patients required specialist referrals that included the screening and surveillance for potential non-CNS organ system anomalies and associated comorbidities including cardiac, renal, ophthalmologic, auditory, and orthopedic disturbances.

## 3. Discussion

Despite a workup that includes electroencephalography (EEG) and brain imaging studies, a substantial number of pediatric epilepsies remain unexplained [23], with up to 67% etiologically classified as unknown or presumed genetic [24]. The growing number of potential etiology-specific therapeutic interventions with accurate diagnosis of an underlying etiology makes NGS an attractive modality for investigating pediatric epilepsies with a high diagnostic yield [25]. This study suggests that NGS can assist in determining the specific etiology of epilepsy in up to 70% of patients with unexplained epilepsy accompanied by neurodevelopmental delays. In this patient population, the number of etiologies altering medical management was substantial, with 1 in 7 patients benefitting from etiology-specific therapeutic changes, and around 50% referred to specialists for additional medical care.

This study provides insights into a population of patients with unexplained epilepsy and with a high NGS diagnostic yield. Indeed, the diagnostic yield with NGS in patients with refractory or non-refractory epilepsy and with neurodevelopmental delays has reached 70–72%. A high rate of consanguinity in the Lebanese population has contributed to this high number. Of note is that the diagnostic yield was also elevated to 61.5% in non-consanguineous families. This diagnostic rate exceeded NGS yields in previous meta-analyses and other patient cohorts with unexplained epilepsy, intellectual disability, or neurodevelopmental disorders, which ranged between 25 and 50% [14,26,27,28]. Added value were the potential changes in life-altering medical management in 13% of the cases. Additionally, families with identified gene mutations benefitted from informed genetic counseling and referral to cardiology, nephrology, ophthalmology, ENT, and orthopedic surgery as needed. Counseling for future pregnancies was provided to 51% of families identified by NGS as having an inherited disease. This study suggests that pediatric epileptologists should consider NGS testing in children with unexplained epilepsy and neurodevelopmental delays, given the high diagnostic yield in this patient group with a likelihood of beneficial altering of management based on the specific genetic defect.

Our data underscore that the relatively narrow phenotypic grouping of the patient population in this cohort led to a high diagnostic yield. The presence of neurodevelopmental disorders improves the yield of genetic testing [5]; hence, wide variabilities in diagnostic rates in the literature may be, at least in part, due to less selectivity in the patient population phenotype that may encompass a spectrum of other comorbidities including milder learning and attention disorders. The children with reported delays in our cohort had significant delays in reaching developmental milestones in at least one domain, but most in all domains. Moreover, while our study was not sufficiently robust to analyze potential correlations between the age of seizure onset and the diagnostic yield of NGS, it is noteworthy that all five patients with neonatal-onset seizures in our cohort had diagnostic NGS in line with the emerging literature on the high yield of genetic testing in unexplained neonatal epilepsy [29].

This study is limited by its retrospective and single-center nature. The fact that our cohort was restricted to the Lebanese population may also limit its generalization to a certain degree. The high rate of consanguinity reached 40% and may have contributed to 78% of the NGS diagnostic yield in such families. While the yield of NGS in non-consanguineous families was higher compared to that reported in the existing literature, one cannot exclude remote consanguinity as a probable contributing factor to the high diagnostic yield in unrelated couples. Indeed, remote consanguinity has been documented in the Lebanese population [30]. Medically intractable epilepsy without developmental delays was noted in just three patients precluding a judgement regarding NGS diagnostic rates in this small group.

## 4. Methods

This is a single-center retrospective chart review performed at AUBMC in Lebanon to identify the diagnostic yield of NGS in children with unexplained epilepsy that is accompanied by neurodevelopmental delays and/or is medically intractable. Approval was obtained from the institutional review board (IRB) at the American University of Beirut. Children who presented to the Special Kids Clinic (SKC) between 2014 and 2019 and underwent NGS were identified, and clinical data were probed by searching the available electronic health records. Patients were included if they were 18 years old or younger at the time of the NGS testing and had unexplained medically intractable epilepsy or unexplained epilepsy accompanied by neurodevelopmental delays (defined as the lack of acquisition of age-appropriate motor and/or language developmental milestones). Medically intractable epilepsy was defined as failure to control seizures despite the use of two appropriately chosen ASMs at therapeutic doses. Patients were excluded if their epilepsy was explained by an acquired brain lesion or insult such as traumatic brain injury, central nervous system (CNS) infections, hypoxic ischemic encephalopathy, stroke, intracranial hemorrhage, and tumors. Exclusion criteria also included epilepsy explained by a vascular malformation or a focal cortical dysplasia. Children who underwent genetic testing prior to NGS were excluded.

Collected data included patient demographics, age at seizure onset, other medical conditions, neurodevelopmental history, seizure frequency, number of ASMs, and EEG and brain magnetic resonance imaging (MRI) results. The rate of genetic testing leading to a change in management recommendations was also reviewed and included initiating or discontinuing specific ASMs or dietary therapies and referral to additional specialists to assess possible non-CNS organ system comorbidities.

Testing and analyses of raw NGS data were performed by the testing company (Centogene, Rostock, Germany). Detected variants were classified based on the recommendations of the American College of Medical Genetics and Genomics (ACMG) as a pathogenic variant (mutation in a gene associated with the patient’s phenotype that has been previously reported as a disease-associated mutation), likely pathogenic variant (a novel variant that is likely deleterious in a gene previously linked to the phenotype), a VUS (a variant that may or may not change protein structure/function with unclear clinical relevance), likely benign variant, and benign variant. The ordering pediatric neurologist/neurogeneticist made the final assessment as to whether a reported VUS explains the patient’s phenotype, thus establishing the presence or absence of a genetic diagnosis.

## 5. Conclusions

This study strongly suggests that NGS should be considered in children with unexplained epilepsy and neurodevelopmental delays. Not only is the diagnostic rate of NGS high in this particular subgroup of patients with unexplained epilepsy, but also genetic testing may lead to potentially life-altering and highly rewarding medical interventions and etiology-specific treatments that reduce morbidity and mortality. Such therapies may maximize neurodevelopmental potential and result in better seizure control.

## Figures and Tables

**Table 1 ijms-25-09645-t001:** Genetic testing results classified according to the predominant clinical phenotype (* subjects with changes in treatment plan based on genetic testing results).

Development and Epileptic Encephalopathy (DEE)	Primarily Neurodegenerative	Metabolic & Neurodegenerative	Others
*CYFIP2*, DEE 65	*TPP1*, neuronal ceroid lipofuscinosis type 2 (n = 2)	*NARS2*, combined oxidative phosphorylation deficiency 24	*IFIH1*, Aicardi–Goutieres disease syndrome 7
*PCDH19*, DEE 9 (n = 2)	*CLN6*, neuronal ceroid lipofuscinosis type 6	*SERAC1*, MEGDEL syndrome	*CHRNE*, slow-channel congenital myasthenic syndrome type 4A
*CACNA1A*, DEE 42 (n = 2)	*AP4M1*, spastic paraplegia 50	*HIBCH*, 3-hydroxyisobutryl-CoA hydrolase deficiency	*TUBA1A*, lissencephaly 3
*GRIN2B*, DEE 27 *	*TSEN54*, pontocerebellar hypoplasia	*QDPR*, BH4-deficient hyperphenylalaninemia type C *	*GLRA1*, hyperkeplexia type 1
*ARV1*, DEE 38	*EIF2B1*, leukoencephalopathy with vanishing white matter	*FOLR1*, neurodegeneration due to cerebral folate transport deficiency *	*SCN1A*, generalized epilepsy with febrile seizure plus *
*SCN8A*, DEE 13 * (n = 2)	Late-infantile neuronal ceroid-lipofuscinoses (CLN 2)		
*PACS2*, DEE 66	*SPAST*, spastic paraplegia 4		
*ATP1A2*, DEE 98	*HEXB*, Sandhoff disease		
	*KCTD7*, progressive myoclonic epilepsy type 3		

**Table 2 ijms-25-09645-t002:** Affected genes in diagnosed patients listed in an alphabetical order along with the coordinates of the identified DNA mutation followed by the resulting amino acid change (mentioned in parentheses).

Gene	Codon>DNA Base, Protein (Amino Acid Change)
*AP4M1*	c.1321C>T, p.(Arg441 *)
*ARV1*	c.294+1G>A
*ATP1A2*	c.160C>T p.(Gln54 *)
*CACNA1A*	c.4526T>C, p.(Phe1509Ser)
*CACNA1A*	c.5018T>C, p.(Leu1673Pro)
*CHRNE*	c.1052C>G, p.(Pro351Arg)
*CLN6*	c.662A>C, p.(Tyr221Ser)
*CLN6*	c.794_796del, p.(Ser265del)
*CYFIP2*	c.3282+858A>G
*EIF2B1*	c.878C>T, (p.Pro293Leu)
*FOLR1*	c.148G>A, p.(Glu50Lys)
*GLRA1*	c.994G>A, p.(Val332Ile)
*GRIN2B*	c.2453T>C, p.(Met818Thr)
*HEXB*	c.1082+5G>A
*HIBCH*	c.452C>T, p.(Ser151Leu)
*IFIH1*	c.500T>G, p.(Leu167Arg)
*KCTD7*	c.509T>C, p.(Ile170Thr)
*NARS2*	c.500A>G, p.(His167Arg)
*PACS2*	c.2588T>C p.(Met863Thr)
*PCDH19*	c.2159C>T, p.(Thr720Ile)
*PCDH19*	c.2656C>T, p.(Arg886 *)
*QDPR*	c.197A>G, p.(Gln66Arg)
*SCN1A*	c.995A>T, p.(Asp332Val)
*SCN8A*	c.2985C>A, p.(Asn995Lys)
*SCN8A*	c.3502C>T, p.(Arg1168Trp)
*SERAC1*	c.1609T>C, p.(Ser537Pro)
*SPAST*	c.1253_1255delAAG, p.(Glu418del)
*TPP1*	c.225A>G, p.(Gln75Gln)
*TPP1*	c.225A>G, p.(Gln75Gln)
*TSEN54*	c.919G>T p.(Ala307Ser)
*TUBA1A*	c.652G>A, p.(Asp218Asn)

**Table 3 ijms-25-09645-t003:** Summary of etiology-specific alterations in treatment plans (ASM: anti-seizure medication; CBZ: carbamazepine; LMT: lamotrigine).

Affected Genes	Resulting Condition	Change in Treatment Plan
*FOLR1*	Neurodegeneration due to cerebral folate transport deficiency	Folinic acid added to regimen
*QDPR*	BH4-deficient hyperphenylalaninemia type C	Tetrahydrobiopterin, folinic acid, and L-dopa added to regimen
*GRIN2B*	Developmental and epileptic encephalopathy type 27	Memantine added to regimen
*SCN1A*	Infantile epileptic encephalopathy type 6 (Dravet syndrome)	LMT switched to another ASM
*SCN8A* (two patients)	Developmental and epileptic encephalopathy type 13	ASM switched to CBZ

## Data Availability

The datasets generated during and/or analyzed during the current study are available from the corresponding author on reasonable request.

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
