# Peer review of "High Diagnostic Yield and Clinical Utility of Next-Generation Sequencing in Children with Epilepsy and Neurodevelopmental Delays: A Retrospective Study"

_ijms, 2024, doi:10.3390/ijms25179645_

Round 1

Reviewer 1 Report

Comments and Suggestions for Authors

The manuscript "High Diagnostic Yield and Clinical Utility of Next-Generation Sequencing in Children with Epilepsy and Neurodevelopmental Delays: A Retrospective Study" presents a valuable investigation of the diagnostic utility of targeted next-generation sequencing (NGS) in a cohort of patients. The authors demonstrate the clinical impact of the technology through high diagnostic yield and provide a clear overview of the study objectives, design, and results. Particularly noteworthy is the focus on changes in treatment plans as a result of the identification of mutations. The focus on treatment plan changes based on identified mutations, resulting from knowledge of the underlying mutations in 6 patients, is particularly noteworthy. The authors also consider the aspect of consanguinity in the patient history. The work certainly conveys original and potentially useful information, but certain improvements are needed to make this report comprehensive. 

Important:

- The authors should provide the exact mutations for all genes listed in Table 1. This is essential to assess the accuracy of the diagnoses and to provide sufficient detail for the reader.

- For each case in Table 1, it is crucial to report both the diagnoses that resulted from the NGS analysis and the diagnoses that were made prior to the NGS test. This will clarify how the patient's condition influenced the decision to perform NGS.

- Authors must provide specific references to support the scientific basis for any changes in the treatment plan. In addition, for all genes involved in treatment changes, a detailed description of their functions and how they relate to the chosen etiologic-based treatment should be included, similar to the approach taken for FOLR1 and QDPR.

- The description of the treatment plan should be expanded to include a brief explanation of the function of the affected gene, the exact dosage administered, and avoid generic terms such as "non-sodium channel blocker ASM".

Minor:

- Given the challenges faced in developing countries which are mentioned in the manuscript, the authors might consider discussing the potential benefits of using a cheaper alternative to WGS/WES, such as targeted NGS of a gene panel that could be suggested based on the initial diagnosis. 

- Line 65: typo "my"

Author Response

Comment 1- The authors should provide the exact mutations for all genes listed in Table 1. This is essential to assess the accuracy of the diagnoses and to provide sufficient detail for the reader.

Response 1: We thank the reviewer for pointing out the importance of including these key details. We certainly agree that reporting the exact coordinates of the mutations enriches the manuscript and improves our communication with the scientific community. In this revised manuscript we have included a new table (Table 2 in the revised version) that summarizes the exact coordinates of the gene mutations of our 31 diagnosed cases.

Comment 2- For each case in Table 1, it is crucial to report both the diagnoses that resulted from the NGS analysis and the diagnoses that were made prior to the NGS test. This will clarify how the patient's condition influenced the decision to perform NGS.

Response 2: We agree with the reviewer that listing pre and post-testing diagnoses is helpful in clarifying the impact of the testing on diagnosis and management. However, the pre-genetic testing our cases were simply labeled as “unexplained epilepsy” with standard descriptions such as focal or generalized that are not etiologically relevant. While some cases appeared to have hints to certain disease categories based on MRI, there were no specific diagnoses made prior to the genetic testing beyond “possible white matter disease” or “epileptic encephalopathy” in few cases.

Comment 3- Authors must provide specific references to support the scientific basis for any changes in the treatment plan. In addition, for all genes involved in treatment changes, a detailed description of their functions and how they relate to the chosen etiologic-based treatment should be included, similar to the approach taken for FOLR1 and QDPR.

Response 3: We agree with the reviewer that these important details should be included. As recommended, in the revised version of this manuscript, we clarified that GRIN2B encodes a subunit of the NMDA receptor and that there is an emerging literature on the benefits of using memantine, an NMDA blocker, in patients with gain of function mutations in GRIN2B along with two new references were added to support these statements (lines 125-130 in the revised manuscript). We also clarified that loss of function in the voltage gated sodium channels secondary to SCN1A mutation prompted a switch in medications from Lamotrigine, a sodium channel blocker to levetiracetam (lines 132-137). We also added a supportive reference (that was mentioned also in the introduction- line 46- to support the background on etiology specific treatments). The same approach was applied to the SCN8A mutation discussions with supportive reference on approaching gain of function in the sodium channel with an anti-seizure medication that blocks this channel (carbamazepine) (lines 138-143). We have also expanded and detailed the treatment approach to QDPR deficiency (lines 120-123 and second row of table 3 in the revised manuscript).    

Comment 4- The description of the treatment plan should be expanded to include a brief explanation of the function of the affected gene, the exact dosage administered, and avoid generic terms such as "non-sodium channel blocker ASM".

Response 4: In this revised manuscript, we clarified the function of the genes that prompted medication changes, explained the choice of the anti-seizure medications, and included the dosages. We clarified the fact that SCN1A -related loss of function in sodium channel mutation prompted a switch from the sodium channel blocker, lamotrigine, to levetiracetam (60 mg/kg/day) and ethosuximide (20 mg/kg/day) (lines 132-137 in the revised manuscript). We also clarified that the SCN8A gain of function in sodium channels prompted the use of the sodium channel blocker carbamazepine with its dose (30 mg/kg/day) (lines 138-143 in the revised manuscript). The treatment plan for QDPR deficiency was also detailed with dosages (lines 120-123 in the revised manuscript).   

Minor:

- Given the challenges faced in developing countries which are mentioned in the manuscript, the authors might consider discussing the potential benefits of using a cheaper alternative to WGS/WES, such as targeted NGS of a gene panel that could be suggested based on the initial diagnosis. 

Response: This is a very interesting point. While more studies are needed, our experience in Lebanon with genetic testing costs encouraged us to avoid sequential testing with individual panels as cumulative costs may be much more costly than the often discounted WES trio and WGS offered by some company.  

- Line 65: typo "my"

Response: “my” was switched to “might” (line 68 in the revised manuscript).

Reviewer 2 Report

Comments and Suggestions for Authors

Dear authors, congratulations on your work. Indeed, NGS can offer significant information for patient management and family counseling. The results of your study are interesting and relevant for infants and children diagnosed with intractable epilepsies with/without neurodevelopmental outcomes. However, please consider the following observations:

-        Re-order the paper sections: material and methods, results, and discussions; a short, relevant conclusion can be drawn based on your results at the end.

-        line 73: Could you detail ”neurodevelopmental delays”? ”medically intractable epilepsy” is explained in the Methods section

-        line 77: ” Seventeen patients bore heterozygous autosomal dominant defects, and 14 others carried homozygous autosomal recessive mutations.” – what do you mean by ”bore

-        line 82: ”The age at seizure onset did not correlate with the diagnostic yield.” – please explain why or how

-        line 109: ”Another patient with a GRIN2B gene defect was started on memantine but the family was noncompliant with therapy.” – what was the outcome of this patient?

-        line 112: ”In one patient, an SCN1A mutation prompted switching lamotrigine to a non-sodium channel blocker ASM. Two other patients with SCN8A defects were started on carbamazepine.” – what was the outcome of these patients?

-        line 131: ”This study suggests that determining the specific etiology of epilepsy through NGS is highly likely when there are neurodevelopmental delays with diagnostic yield around 70%. – please re-phrase for better understanding

        Could you please comment on the 3 likely pathogenic variants identified in your study?

Comments on the Quality of English Language

There are some minor English language errors to correct

Author Response

- Comment 1: Re-order the paper sections: material and methods, results, and discussions; a short, relevant conclusion can be drawn based on your results at the end.

Response 1: Thank you for this helpful suggestion. As recommended, the conclusions section was moved to the end of the manuscript. We will gladly re-order the other sections if the editor and journal allow it (the Methods sections has to be after the discussion based on the journal’s guide for authors). 

- Comment 2:  line 73: Could you detail” neurodevelopmental delays”?” medically intractable epilepsy” is explained in the Methods section

Response 2: We thank the reviewer for pointing out this omission. In this revised manuscript, we clarified in the method section that neurodevelopmental delays refer to not reaching age-appropriate motor and/or language developmental milestones within the expected timeframes (lines 217-218 in the revised version of the manuscript).

- Comment 3: line 77:” Seventeen patients bore heterozygous autosomal dominant defects, and 14 others carried homozygous autosomal recessive mutations.” – what do you mean by” bore”

Response 3: Since the use of “bore” can be confusing, it was changed to “carried” (line 84 in the revised version of the manuscript).

- Comment 4:  line 82:” The age at seizure onset did not correlate with the diagnostic yield.” – please explain why or how

Response 4: We agree that this statement is vague and cannot be supported by robust statistical data since this work is not powered enough for such correlative analyses. However, we noticed that there were only 5 patients with neonatal onset seizures, and they all tested positive. We therefore included this finding in the result section (line 89 in the revised manuscript) and reworded the statement to: “while our study was not powered enough to analyze potential correlations between age of seizure onset and the diagnostic yield of NGS, it is noteworthy that all 5 patients with neonatal-onset seizures in our cohort had a diagnostic NGS in line with the emerging literature on the high yield of genetic testing in unexplained neonatal epilepsy” (a supportive reference was also added) (lines in the revised manuscript) (lines 192-195 in the revised manuscript).      

- Comment 5:   line 109:” Another patient with a GRIN2B gene defect was started on memantine but the family was noncompliant with therapy.” – what was the outcome of this patient?

Response 5: Thank you for pointing this omission. As recommended, we clarified that that this child did not experience a change in clinical status (lines 128-130 in the revised manuscript).

- Comment 6: line 112:” In one patient, an SCN1A mutation prompted switching lamotrigine to a non-sodium channel blocker ASM. Two other patients with SCN8A defects were started on carbamazepine.” – what was the outcome of these patients?

Response 6: Thank you for pointing out this omission. In the revised manuscript, we added details about the decrease in seizure frequency in the SCN1A patient (seizure free for 18 months at last follow up) (line 137 in the revised manuscript), and the SCN8A patients (6 months seizure free at last follow up in one, and a drop from multiple daily seizures to weekly seizures in the other) (lines 141-143 in the revised manuscript).

- Comment 7: line 131:” This study suggests that determining the specific etiology of epilepsy through NGS is highly likely when there are neurodevelopmental delays with diagnostic yield around 70%. – please re-phrase for better understanding

Response 7: We agree that the writing is difficult to understand. As recommended, we rephased to: “This study suggests that NGS can assist in determining the specific etiology of epilepsy in up to 70% of patients with unexplained epilepsy accompanied by neurodevelopmental delays.” (lines 160-163 in the revised manuscript).

- Comment 8: Could you please comment on the 3 likely pathogenic variants identified in your study?

Response 8: As requested, we clarified that 3 genes (TUBA1A, AP4M1, ATP1A2) were classified as likely pathogenic by the testing company and based on ACMG guidelines. We also clarified that these were deemed diagnostic by the ordering pediatric neurologist/neurogeneticist (lines 82-83 in the revised manuscript).